# LOAD BALANCING MIXTURE OF EXPERTS WITH SIMILARITY PRESERVING ROUTERS

## ABSTRACT

Sparse Mixture of Experts (MoE) models offer a scalable and efficient architecture for training large neural networks by activating only a subset of parameters ("experts") for each input. A learned router computes a distribution over these experts, and assigns input tokens to a small subset. However, without auxiliary balancing mechanisms, routers often converge to using only a few experts, severely limiting model capacity and degrading performance. Most current load balancing mechanisms encourage a distribution over experts that resembles a roughly uniform distribution of experts per token. During training, this can result in inconsistent routing behavior, resulting in the model spending its capacity to learn redundant knowledge. We address this by introducing a novel load balancing loss that preserves token-wise relational structure, encouraging consistent expert choices for similar inputs during training. Our experimental results show that applying our loss to the router results in 36% faster convergence and lower redundancy compared to a popular load balancing loss.

## 1 INTRODUCTION

As the demand for larger and more capable neural networks continues to grow (Kaplan et al., 2020; Brown et al., 2020), the need for architectures that can scale efficiently—without incurring prohibitive computational costs—has become increasingly important. This is especially true in the context of large language models (LLMs), where state-of-the-art performance often requires billions of parameters and massive training datasets. One such approach, the Mixture of Experts (MoE) model (Shazeer et al., 2017), introduces sparsely activated sub-networks at certain layers, allowing for increased model capacity while preserving computational efficiency.

While MoE architectures offer improved parameter scalability, they often suffer from poor expert utilization during pretraining. Without mechanisms that encourage balanced routing, the model frequently learns to rely on only a small subset of experts (Eigen et al., 2014; Bengio et al., 2016). Typically, routing decisions are made per token using a learned router that outputs a probability distribution over experts—a paradigm known as Token Choice (TC) (Fedus et al., 2022). To encourage balanced expert usage, various strategies have been proposed, including sequence-level auxiliary losses such as load balancing loss (LBL) (Fedus et al., 2022) or the Expert Choice (EC) routing variant which generates a distribution over a sparse set of activated tokens for each expert (Zhou et al., 2022). Section 5 covers additional strategies for load balancing.

Load balancing strategies often encourage a uniform distribution over experts to avoid collapse. This approach has proven to be useful to stabilize MoEs during training, and has been used in many recent works (Muennighoff et al., 2025; Dai et al., 2024; DeepSeek-AI et al., 2025; Xue et al., 2024). However, in this paper, we argue that imposing a uniform distribution over experts causes MoE models to expend their capacity acquiring the same knowledge across multiple experts. Besides the inefficiencies imposed by this approach, exposing similar tokens to several different experts during training results in inconsistent routing behavior and expert assignments. This in turn further exacerbates knowledge redundancy across experts. Previous work (Dai et al., 2024; Liu et al., 2024) suggests that the amount of knowledge shared between experts is correlated to losses in performance.

To encourage consistent expert assignments for similar input tokens during training, we propose preserving the relational structure among tokens during routing, resulting in similar expert distributions for similar tokens. We achieve this by promoting orthogonality in the router's weights, as orthogonal

matrices are dot-product (and thus, angle) preserving. We introduce **sim**ilarity-preserving routers for MoE load **bal**ancing (SIMBAL), a novel load balancing auxiliary loss that maintains token-wise relational structure by softly encouraging orthogonality in the router weights. Unlike methods that impose orthogonality through explicit parameter constraints—which are computationally expensive and numerically unstable (see Section 4.1)—SIMBAL aligns the Gram matrix ($Q^\top Q$) of router weights with the identity matrix. This softly regularizes router outputs to preserve pairwise token similarities, achieving the benefits of orthogonal routing with significantly lower computational cost.

By maintaining semantic structure and promoting diverse expert usage, SIMBAL reduces redundancy, accelerates convergence, and improves final model quality. Our models require 36% fewer tokens when training to achieve the same loss as LBL, and achieve 0.213 lower perplexity given the same compute budget.

## 2 BACKGROUND

### 2.1 MIXTURES OF EXPERTS

A Mixture of Experts (MoE) model *sparsely activates* certain parameters during inference, in contrast to standard dense networks where all parameters are used. In this work, we focus on Mixture of Experts models for the Transformer architecture (Vaswani et al., 2017), a popular choice for training models on sequence-wise data such as those seen in natural language.

Transformers are typically composed of a series of blocks, each consisting of a self-attention module followed by a feed-forward network (FFN). The FFN is usually a two-layer fully connected network with a large hidden dimensionality. For example, given an input vector $x \in \mathbb{R}^{D_M}$, where $D_M$ is the model (input/output) dimensionality, the standard FFN computes:

$$\text{FFN}(x) = W_2 \cdot \sigma(W_1 x + b_1) + b_2, \tag{1}$$

where $W_1 \in \mathbb{R}^{D_F \times D_M}$, $W_2 \in \mathbb{R}^{D_M \times D_F}$, $b_1 \in \mathbb{R}^{D_F}$, and $b_2 \in \mathbb{R}^{D_M}$. The intermediate hidden dimension $D_F$ is typically much larger than $D_M$. The nonlinearity $\sigma$ is an activation function; we use SwiGLU (Shazeer, 2020).

In a Mixture of Experts Transformer, the FFN is replaced by a set of smaller, parallel FFNs called "experts." Let there be $E$ such experts. Each expert has its own parameters $\{W_1^{(e)}, W_2^{(e)}, b_1^{(e)}, b_2^{(e)}\}$, where $W_1^{(e)} \in \mathbb{R}^{D_E \times D_M}$, $W_2^{(e)} \in \mathbb{R}^{D_M \times D_E}$, $b_1^{(e)} \in \mathbb{R}^{D_E}$, and $b_2^{(e)} \in \mathbb{R}^{D_M}$. Here, $D_E$ is the hidden dimension used within each expert.

A routing mechanism assigns each token $x \in \mathbb{R}^{D_M}$ to a small subset of $A$ activated experts (typically $A \ll E$). The router is a linear transformation $R \in \mathbb{R}^{D_M \times E}$ followed by a sparse top-$A$ selection, producing expert indices $i_1, \ldots, i_A$ and associated routing weights $r_1, \ldots, r_A$. The MoE layer then computes:

$$\text{MoE}(x) = \sum_{a=1}^{A} r_a \cdot \left( W_2^{(i_a)} \cdot \sigma(W_1^{(i_a)} x + b_1^{(i_a)}) + b_2^{(i_a)} \right). \tag{2}$$

This definition of the MoE can also be viewed as a weighted sum over expert FFN outputs, skipping the computation for any expert where the weight is zero. This architecture enables scaling model capacity via $E$ without a proportional increase in computational cost, as only $A$ experts are active per input.

### 2.2 EXPERT ROUTING

Despite the small parameter count of MoE routers (in our larger setting, 0.018% of the total parameters), they have an outsized impact on the performance and capacity of the model, as they orchestrate billions of parameters. Thus, it is imperative to pay careful attention to this mechanism when training MoE models. In MoE Transformers, routing is computed from the previous attention output $x \in \mathbb{R}^{D_M}$ via a learned router matrix $R \in \mathbb{R}^{D_M \times E}$, producing scores $xR \in \mathbb{R}^E$. Applying a gating function $G$

results in routing weights $r = G(xR)$. We use softmax, which generates a probability distribution over experts, from which the top-$A$ active experts are selected and weighted for each token.

We compare our approach to balancing with the Load Balancing Loss (LBL) presented by Fedus et al. (2022). This setup is highly popular and represents the state-of-the-art, being used in Muennighoff et al. (2025); DeepSeek-AI et al. (2025); Dai et al. (2024), and (Xue et al., 2024) (we give an overview of alternative methods and their limitations in Section 5.) LBL encourages uniform expert usage by correlating how frequently each expert is selected with how much routing weight it receives. Let $f_i$ be the fraction of tokens routed to expert $i$, $P_i$ the average routing probability for expert $i$, and $E$ the number of experts. The LBL is defined as:

$$\mathcal{L}_{\text{LBL}} = \alpha \cdot E \cdot \sum_{i=1}^{E} f_i \cdot P_i \tag{3}$$

Minimizing this loss encourages the router to distribute tokens more evenly across experts. However, it may require tuning of a loss coefficient $\alpha$ to avoid overpowering the main training objective. We include PyTorch implementation details in Appendix A.4.

## 3 METHODS

We propose preserving token-wise structural relationships to ensure effective and consistent usage of experts during training. We accomplish this by encouraging orthogonality in the router, which preserves the pairwise angles of the inputs. In this section, we explain the methods used to achieve our results, and our design choices.

### 3.1 LOAD BALANCING VIA ORTHONORMAL ROUTERS

A natural strategy to ensure expert choices correlate with token-wise relationships is to constrain the router weights to form an orthonormal (and thus, dot-product preserving) matrix. PyTorch (Paszke et al., 2019) provides a utility for this using a QR decomposition, producing a matrix $Q \in \mathbb{R}^{m \times n}$ such that $Q^\top Q = I_n$ if $m \geqslant n$ (as is typically the case with MoE routers).

While appealing, the cost of using this orthogonal parameterization is prohibitively expensive in wall-clock time when applied to large-scale models, because the algorithms used to ensure this property are computationally expensive. Instead, we propose a loss that encourages structure preservation without requiring explicit parameterization.

Let the router be a matrix $R \in \mathbb{R}^{D_M \times E}$, where $D_M$ is the model dimension and $E$ is the number of experts. Since $E \ll D_M$, we minimize the deviation of the Gram matrix $R^\top R$ from the identity:

$$\mathcal{L}_{\text{orth}} = \left\| R^\top R - I_E \right\|_1 \tag{4}$$

This loss is dataset-agnostic and computationally cheap. This is important, as Qiu et al. (2025) finds that existing losses, which are dependent on the data, require large batch sizes to be effective. We additionally initialize the router with a (near) orthogonal initialization (Saxe et al., 2014) (though it should be sufficient to simply run a few router-only training steps, see Table 2), as we find it results in quicker convergence. We call this method SIMBAL, as we are effectively balancing by preserving the pair-wise similarity of the tokens. The experiments in our paper scale this coefficient by 0.1, but we find that this is not important, as shown in Section 4.2. We include PyTorch implementation details in Appendix A.4.

### 3.2 MODEL ARCHITECTURE AND TRAINING

**Model Architecture.** Our model architecture closely follows prior work by OLMo et al. (2025) and Muennighoff et al. (2025). We use a Transformer backbone with RMSNorm (Zhang and Sennrich, 2019), SwiGLU activations (Shazeer, 2020), and Rotary Position Embeddings (RoPE) (Su et al., 2021). We apply Z-loss Team (2025); Chowdhery et al. (2022) with a coefficient of 1e-5, as in OLMo et al. (2025). Unlike OLMo 2, we do not modify the placement of normalization layers nor do

Table 1: Parameters used for the model architecture and training. Parameter (active, total) counts include token embeddings. All MoE models have 32 experts, with the top 4 activated.

| Parameter | Dense-M | MoE-M | Dense-L | MoE-L |
|---|---|---|---|---|
| $D_M$ | 768 | 768 | 1536 | 1536 |
| Depth | 8 | 8 | 12 | 12 |
| Heads | 8 | 8 | 12 | 12 |
| $D_F$ | 3072 | 768 | 6144 | 1536 |
| RoPE $\theta$ | 1e4 | 1e4 | 1e5 | 1e5 |
| Peak LR | 5e-4 | 5e-4 | 3e-4 | 3e-4 |
| Embedding Params | 77M | 77M | 154M | 154M |
| Active Params | 230M | 230M | 761M | 761M |
| Total Params | 230M | 627M | 761M | 3.14B |

we apply QK-Norm (Dehghani et al., 2023). We replace all FFN layers with MoE layers. Further architectural details can be found in Table 1. Our implementation largely builds upon the open-source OLMo codebase (OLMo et al., 2025), except for data loading and processing due to differences in infrastructure. For the LBL baseline, we follow Muennighoff et al. (2025) and Wang et al. (2024), using a loss coefficient of 0.01.

**Model Scales and Training.** We pretrain models at two scales: a medium model (MoE-M) with 230M active and 627M total parameters, and a large model (MoE-L) with 762M active and 3.14B total parameters (including embeddings). For each scale, we performed a brief hyperparameter sweep across three learning rates. All models are trained using the AdamW optimizer (Loshchilov and Hutter, 2019), with a weight decay of 0.01, linear warm-up from 10% of the peak learning rate over 2000 steps, followed by cosine decay (Loshchilov and Hutter, 2017) to 10% of the peak learning rate. Additional model specifications are listed in Table 1. All model parameters are in `bfloat16`.

All models are trained on a subset of tokens from the DCLM-pool-400m-1x dataset (Li et al., 2025) (used in other work such as Muennighoff et al. (2025)), tokenized using the cl100k_base tokenizer from the tiktoken library (OpenAI, 2024). We reserve one file shard (77M tokens) for validation. All MoE-M models are trained on 19.9B tokens, while MoE-L mdoels are trained on 78.6B tokens. No further fine-tuning is performed, as our focus is on the pretraining phase, which is typically the most computationally intensive stage of LLM development.

**Compute and FLOP Estimates.** All models are trained using Distributed Data Parallelism (DDP) (Li et al., 2020). For MoE-M, we use 8 NVIDIA A100 40GB GPUs per training run; for MoE-L, we use 8 AMD MI300X 192GB accelerators.

To estimate total training FLOPs, we follow the approximation from Brown et al. (2020), using $6 \times N \times T$ per forward pass, where $N$ is the number of non-embedding active parameters and $T$ is the number of training tokens.

For MoE-M and Dense-M, with 230M active parameters and 77M in embeddings, trained on $2 \times 10^{10}$ tokens, this results in:

$$6 \times ((230 - 77) \times 10^6) \times 2 \times 10^{10} = \mathbf{1.836 \times 10^{19}} \text{ FLOPs}$$

For MoE-L and Dense-L, with 761M active parameters and 154M in embeddings, trained on $7.8 \times 10^{10}$ tokens, this results in:

$$6 \times ((761 - 154) \times 10^6) \times 7.8 \times 10^{10} = \mathbf{2.840 \times 10^{20}} \text{ FLOPs}$$

### 3.3 MEASURING EXPERT SIMILARITY

Previous work measures expert specialization by dropping the top fraction of experts and recording the resulting performance degradation (Dai et al., 2024). To identify redundant experts, including those no longer selected once routing saturates, we would need to drop progressively more experts beyond the top $K - 1$ experts and measure which experts with lower weights minimally degrade performance. Doing this at each MoE layer requires many separate ablations, and because deeper-layer expert usage depends on earlier-layer outputs, these evaluations cannot be reused across layers.

Table 2: Comparison of orthogonality preservation methods, average and standard deviation over 100 trials. We report the maximum deviation from orthonormality (**Max Dev**) and the mean L1 distance to the identity matrix (**L1 Dist**) after casting to our training precision. **Trained** refers to our loss-based method after 100 optimization steps. **Param** uses the orthogonal parameterization from Lezcano-Casado (2019). **OrthoInit** follows the initialization from Saxe et al. (2014). All matrices have shape $1536 \times 32$, matching our router dimensions. Best results in each column are bolded.

| Method | Max Dev | L1 Dist |
|---|---|---|
| Trained | $\mathbf{1.03 \times 10^{-5} \pm 2.76 \times 10^{-6}}$ | $\mathbf{8.52 \times 10^{-7} \pm 5.84 \times 10^{-8}}$ |
| Param | $2.00 \times 10^{-4} \pm 2.31 \times 10^{-5}$ | $4.80 \times 10^{-5} \pm 1.60 \times 10^{-6}$ |
| OrthoInit | $1.93 \times 10^{-4} \pm 1.88 \times 10^{-5}$ | $4.62 \times 10^{-5} \pm 1.79 \times 10^{-6}$ |

As a result, the total cost scales with both the number of experts and the number of layers, typically requiring hundreds of full-model evaluations per token. This makes expert dropping prohibitively expensive for regular use and limits it to occasional, large-scale analyses.

To enable finer-grained and more frequent monitoring of specialization, we introduce *Pairwise Expert Similarity (PES)*, a smoother, scalable, and more computationally efficient metric. PES directly measures how similar experts' outputs are on a shared batch of tokens, avoiding the need for repeated ablations. Crucially, computing PES requires only one additional inference pass in which all experts in a layer are evaluated once. In our models, this cost corresponds to a 3.6 to 4.9 times FLOP multiplier per token, which is small enough to run periodically during development without meaningfully affecting training or evaluation cost. By contrast, the cost of expert dropping prevents it from being applied at similar frequency. PES is defined as:

$$\text{PES}_{\text{model}} = \frac{1}{|B|} \sum_{b \in B} \mathcal{C}_{\text{expert}}(\mathbf{x}) \tag{5}$$

$$\mathcal{C}_{\text{expert}}(\mathbf{x}) = \frac{2}{N(N-1)} \sum_{i=1}^{N} \sum_{j=i+1}^{N} \cos\left(\mathbf{f}_i(\mathbf{x}), \mathbf{f}_j(\mathbf{x})\right) \tag{6}$$

Here, $\mathcal{C}_{\text{expert}}(\mathbf{x})$ denotes the mean cosine similarity of expert outputs for batch sample $\mathbf{x}$, and $\text{PES}_{\text{model}}$ is the batch-averaged similarity across all $|B|$ samples. $N$ is the number of experts, $\mathbf{f}_i$ is the function computed by the $i$-th expert. The cosine similarity $\cos(\mathbf{u}, \mathbf{v})$ is defined as $\frac{\mathbf{u} \cdot \mathbf{v}}{\|\mathbf{u}\| \cdot \|\mathbf{v}\|}$, measuring the angle between output vectors.

Lower PES values indicate more diverse and less redundant expert behaviors. Because PES evaluates all experts and provides a continuous pairwise measure of functional similarity, it offers a detailed diagnostic of specialization. We compute PES using 4 million randomly sampled tokens.

## 4 EXPERIMENTS

### 4.1 ORTHOGONALIZATION AND BALANCING

Our key contribution is that we perform load balancing by using a router that is encouraged to be orthogonal, and thus preserves token-wise relationships. Rather than enforcing orthogonality through explicit parameter constraints—which is computationally expensive, requires frequent reparameterization, and is prone to numerical instability, particularly when training large-scale models—we instead use the loss function described in Section 3.1. We now evaluate the effectiveness of promoting orthogonality in the router.

As PyTorch currently lacks support for orthogonal parameterizations in lower-precision formats commonly used to train language models (that we use), we perform orthogonalization in `float32`, and then cast the resulting matrix to `bfloat16`, our training precision. Our loss-based method trains the matrix directly in `bfloat16`. We report both the maximum and mean deviation from orthonormality, as well as the final loss values, in Table 2. We find that our loss consistently produces matrices that more closely approximate orthonormality than direct orthogonal parameterizations

Table 3: Load balancing and orthogonalization of LBL and SIMBAL on MoE-L.

| Metric | SEU | Entropy | $(R^T R - I)^2$ |
|--------|-----|---------|------------------|
| LBL | 1.000 | 1.268 | 0.0311 |
| SIMBAL | 0.991 | 1.168 | $2.121 \times 10^{-8}$ |

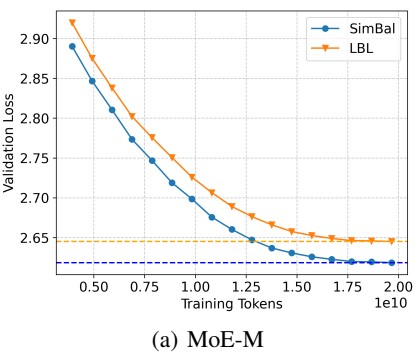

(a) MoE-M

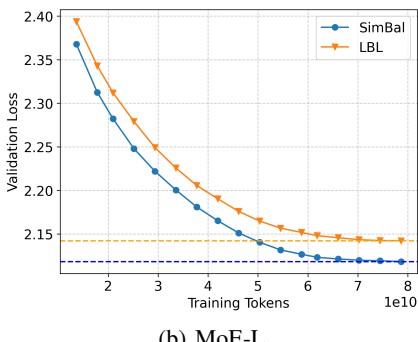

(b) MoE-L

Figure 1: Validation loss curves for checkpoints during training. In both MoE-M and MoE-L, we achieve the same loss roughly 36% faster.

in our scenario. In fact, our approach matches or exceeds the throughput of efficient orthogonal parameterizations, while avoiding the need for expensive reorthogonalization steps. For this synthetic experiment, we train with AdamW (with no weight decay), and a learning rate cosine decayed from $1 \times 10^{-4}$ to $1 \times 10^{-5}$ over 100 consecutive steps. In our MoEs, we simply add our loss as an auxiliary loss term and update once per language model training step. We examine the coefficient sensitivity of SIMBAL to determine if tuning is necessary.

In terms of expert utilization in MoEs, our method avoids collapse comparably to LBL, ensuring that no experts remain unutilized. Figure 6 illustrates the unique expert usage over time at two different scales, compared to LBL and using no losses (which results in unused experts). To verify that sequence-wise balance is not substantially degraded, we compare SIMBAL against LBL by measuring the entropy of the routing distributions and Sequence-wise Expert Utilization (SEU), as reported by the mean over the fraction of experts used per sequence, to show that load balance within a sequence is not significantly degraded. We report our results in Table 3.

To analyze whether SIMBAL is able to effectively orthogonalize routing matrices, we analyze the mean layer-wise L2 distance of the final router gram matrix from the identity matrix in Table 3. More in-depth data with layer-wise values across MoE-L and MoE-M can be found in Appendix A.3.

## 4.2 LANGUAGE MODELING

We compare our method to LBL by training language models according to the setup described in Section 3.2, evaluating performance based on the perplexity of the final checkpoint. The resulting models are reported in Table 4. We additionally report the SEU of the models.

Across both MoE-M and MoE-L scales, SimBal converges approximately 36% faster than LBL. We show validation values during training in Figure 1 For MoE-L, SimBal approaches the target loss after processing roughly 50B tokens, compared to 78.6B for LBL—a 36% improvement. Similarly, in the MoE-M setting, SimBal reaches comparable loss levels at around 12.7B tokens, versus 19.9B for LBL. We additionally evaluate MoE-L on standard downstream benchmarks to test whether the perplexity gains of SIMBAL translate to broader tasks, comparing against LBL (Table 5). Overall, our method outperforms LBL in both downstream performance and training efficiency.

We train 4 additional models (for a total of 5 models) for both SIMBAL and LBL on MoE-M (due to computational limitations) to parse the statistical significance of our results. We find that models trained with LBL have a mean perplexity of 14.051 with a standard deviation of 0.026. In comparison, SIMBAL achieves a mean perplexity of 13.691 with standard deviation 0.039. The mean SIMBAL

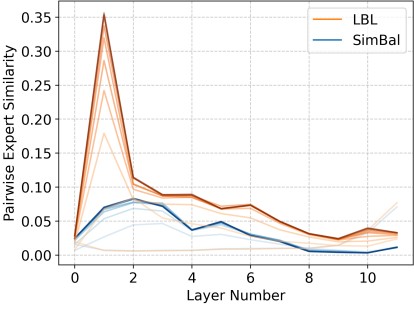

(a) Redundancy Per Layer, Lower = Better

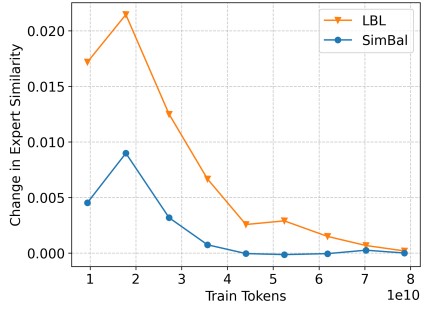

(b) Δ Redundant Expert Knowledge

Figure 2: Analysis of expert redundancy in MoE-L models. **(a)** PES across different layers, our approach (blue) maintains significantly lower redundancy than LBL (orange). Darker = later in training. **(b)** Rate of change of PES during training, averaged over all layers. Redundancy occurs when many distinct experts see similar tokens, and is most likely to happen early in training, as we observe. We note that this is $> 0$ at most points for LBL, suggesting it exacerbates redundancy during the majority of training.

Table 4: Model setup and performance.

| Model | Dense-M | MoE-M | MoE-M | Dense-L | MoE-L | MoE-L |
|---|---|---|---|---|---|---|
| **Balancing** | – | LBL | SimBal | – | LBL | SimBal |
| **Perplexity ↓** | 19.468 | 14.086 | 13.685 | 10.047 | 8.517 | 8.304 |
| **Min PES ↓** | – | 0.0255 | 0.0044 | – | 0.0241 | 0.0028 |

Table 5: Comparison of LBL-L and SimBal-L performance across benchmarks.

| Benchmark | LBL-L $\pm$ stderr | SimBal-L $\pm$ stderr |
|---|---|---|
| ARC Challenge (Clark et al., 2018) | $22.44\% \pm 1.22\%$ | $23.21\% \pm 1.23\%$ |
| ARC Easy (Clark et al., 2018) | $40.49\% \pm 1.01\%$ | $41.16\% \pm 1.01\%$ |
| HellaSwag (Zellers et al., 2019) | $35.45\% \pm 0.48\%$ | $35.74\% \pm 0.48\%$ |
| PIAQ (Bisk et al., 2019) | $66.49\% \pm 1.10\%$ | $66.81\% \pm 1.10\%$ |
| WinoGrande (Sakaguchi et al., 2019) | $49.72\% \pm 1.41\%$ | $52.49\% \pm 1.40\%$ |
| GLUE (Wang et al., 2018) | $45.10\% \pm 1.98\%$ | $51.73\% \pm 1.97\%$ |
| mean | $43.28\%$ | $45.19\%$ |

performance is over 13 standard deviations lower than the perplexity of LBL, showing that our results are very statistically significant.

Finally, we examine sensitivity to the auxiliary loss coefficient (0.01, 0.1, 1.0), with results in Table 7. Based on our 5-seed runs on MoE-M, the effect is negligible, and we do not recommend tuning this hyperparameter.

### 4.3 REDUNDANCY AND SPECIALIZATION IN EXPERTS

Motivated by Dai et al. (2024), we study expert specialization and redundancy. As described in Section 3.3, we measure these properties with Pairwise Expert Similarity (PES), in contrast to their expert dropout approach. In Figure 3, we validate the correlation between PES and their method, reproducing their redundancy analysis. By their metric, SIMBAL shows lower redundancy, as validation perplexity rises more sharply when top experts are dropped. However, such dropout-based metrics lack granularity and are prohibitively expensive for large-scale evaluation. PES instead provides a lightweight, scalable measure of redundancy, enabling per-layer, per-checkpoint analysis across all experts in parallel.

We hypothesize that SIMBAL produces less redundant experts than LBL. LBL enforces uniform distributions, leading to instability in early training as changing embeddings cause frequent routing

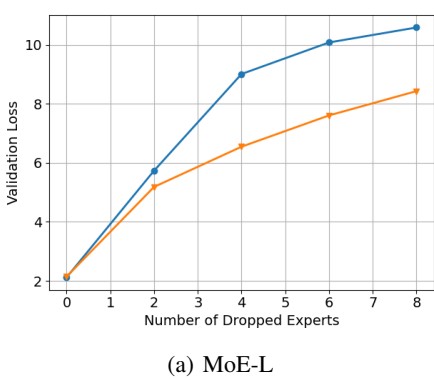

(a) MoE-L

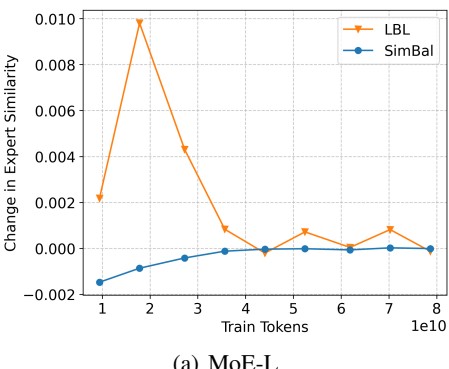

(a) MoE-L

Figure 3: Number of dropped top experts vs. validation loss, as proposed by Dai et al. (2024). SIMBAL exhibits lower redundancy, as shown by larger degradation when top experts are dropped.

Figure 4: Rate of change in minimum PES (over the layers of a model) over a training run, comparing LBL (higher perplexity) and SimBal (lower perplexity).

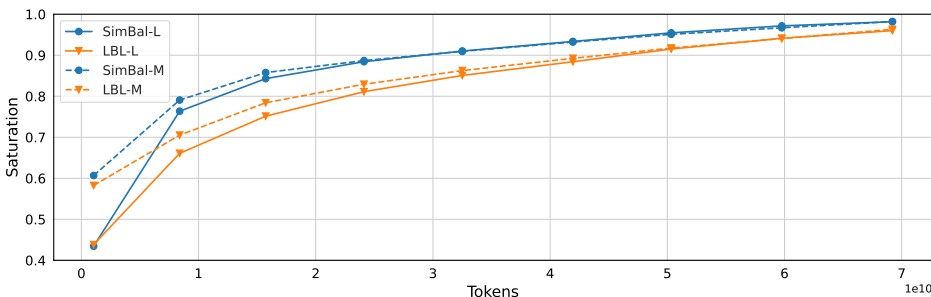

Figure 5: Router saturation curves. SimBal saturates notably faster than LBL.

shifts. Under near-uniform assignment, small input perturbations can reassign tokens, creating redundancy as many experts see similar tokens. We capture this effect by measuring changes in redundancy.

As shown in Figure 2(b), most redundancy in LBL (orange) arises early, coinciding with embedding volatility and unstable routing. Redundancy remains above zero through much of training, reinforcing that LBL amplifies it. In contrast, SIMBAL (blue) stabilizes quickly: while expert distributions adapt, they converge to consistently lower PES (Figure 2(a)). Moreover, the rate of change remains near zero for most of training, showing that our method avoids the issues of LBL. To assess how these effects relate to routing stability, we plot router-saturation curves in Figure 5. Each curve reports the agreement in expert assignments between a given checkpoint and the final model. Results for MoE-M and MoE-L are shown under both SIMBAL and LBL. We observe that SIMBAL saturates more quickly and maintains a consistently higher saturation level across checkpoints at both scales, indicating greater routing stability.

Final PES values are summarized in Table 4. To reduce sensitivity to outliers, we report the minimum PES across all layers, filtering out spikes in a single individual layer (common with LBL). We choose minimum, since we do not observe substantial dips in PES by layer, primarily jumps, and we wanted this metric to be as simple and intuitive as possible. SimBal consistently produces models with substantially lower minimum PES than LBL. Figure 4 shows the rate of change in minimum PES over time.

### 4.4 INFERENCE-TIME EXPERT PRUNING

We further evaluate SIMBAL under inference-time *expert pruning*, following Szatkowski et al. (2024), where experts with assignment probabilities below a threshold are dropped at runtime. Results are

Table 6: Dynamic-K expert stage 3 selection (Szatkowski et al., 2024) synergy with SimBal vs. LBL (perplexity and runtime on a full validation run, MoE-L). SimBal is able to provide similar or better perplexity with lower runtime when properly configured.

| Dropped $P(E) <$ | SimBal (PPL) | SimBal (s) | LBL (PPL) | LBL (s) |
|---|---|---|---|---|
| 0 | 8.304 | 620.927 | 8.517 | 619.657 |
| 0.1 | 8.364 | 571.147 | 8.542 | 575.121 |
| 0.15 | 8.598 | 503.065 | 8.621 | 543.027 |
| 0.2 | 9.380 | 472.915 | 9.057 | 495.200 |

Table 7: Performance across three scaling coefficients to SimBal. We find that the differences are not significant enough to warrant hyperparameter tuning.

| Model | MoE-M | MoE-M | MoE-M |
|---|---|---|---|
| **Coefficient** | 1.0 | 0.1 | 0.01 |
| **Perplexity** ↓ | 13.716 | 13.685 | 13.687 |
| **Min PES** ↓ | 0.0045 | 0.0044 | 0.0050 |

presented in Table 4.4. SimBal produces less uniform assignments, allowing pruning to drastically improve efficiency with minimal perplexity cost. In contrast, LBL shows weaker synergy with pruning: while its performance drop is smaller (likely due to redundancy, similarly to Figure 3(a)), improvements in throughput are limited. Notably, when experts below a weight of $0.15$ are dropped (where both perplexities are most similar), SimBal achieves a 7.4% speedup (543s vs. 503s).

## 5   RELATED WORK

There has been significant interest in MoE models for scaling LLMs, as shown in Lepikhin et al. (2020); Zoph et al. (2022); Fedus et al. (2022); Xue et al. (2024); DeepSeek-AI et al. (2025); Databricks (2024); Llama (2025); Muennighoff et al. (2025), and more. We explore related design choices below.

**Routing and Load Balancing Mechanisms.** Efficient routing in MoE architectures involves selecting appropriate experts for each token (Token Choice) (Fedus et al., 2022) while ensuring balanced expert utilization. Some previous work suggests allowing experts to choose the tokens they process (Expert Choice) (Zhou et al., 2022), but this tends to have issues regarding performance in autoregressive generation (Muennighoff et al., 2025), and leak information about future tokens (Wang et al., 2024).

Traditional approaches employ an auxiliary load balancing loss (Fedus et al., 2022) to encourage a uniform distribution over experts, which can interfere with the main training objective and potentially degrade performance. To address this, auxiliary-loss-free (LF) strategies have been introduced (Wang et al., 2024), notably used in DeepSeek-V3 (DeepSeek-AI et al., 2025), but always in conjunction with an auxiliary balancing loss. LF dynamically adjusts per-expert bias terms added to the routing scores, guiding top-$K$ expert selection without introducing additional gradients. While this improves global balance, it struggles to balance MoE usage *sequence-wise*, often degrading throughput.

Due to difficulties in achieving effective load balance in our early experiments, we did not pursue full-scale MoE-L training with LF in the main paper, and instead provide an in-depth analysis in Appendix A.2. Moreover, LF is highly sensitive to batch size: Qiu et al. (2025) report a substantial perplexity drop when training with batch size $512$ vs. $4$ (per-device, no sync). This effect is far milder for LBL, and entirely absent for SimBal, which is invariant to the data. Finally, while Qiu et al. (2025) argue that LBL requires distributed synchronization to maximize batch size and improve specialization, SimBal eliminates this need altogether.

**Orthogonality in MoE.** Prior studies have applied orthogonality to diversify expert representations in MoE models. OMoE (Liu et al., 2024) introduces an optimizer that updates each expert in a direction orthogonal to the subspace spanned by other experts, enhancing representation diversity. MOORE (Hendawy et al., 2024) employs the Gram-Schmidt process to enforce orthogonality among expert representations in multi-task reinforcement learning. In contrast, our approach applies orthogonality at the *router* level, not the experts themselves. This strategy offers computational efficiency by avoiding expensive operations during training and allows seamless integration into existing architectures.

Moreover, by not constraining expert weights, we avoid potential performance degradation due to restrictive parameter constraints.

Parallel work such as ERNIE 4.5 (Baidu-ERNIE-Team, 2025) introduces a loss related in spirit to ours, though there are several important differences in formulation and effect. Their method normalizes the router weights and measures cosine similarity between the resulting Gram matrix and the identity. Because cosine similarity is scale-invariant, this approach does not encourage $R^\top R \approx I$ and therefore does not produce an orthogonal router; rather, it primarily aims to decorrelate expert assignments. In contrast, our method directly encourages the unnormalized router weights to approach a semi-orthogonal structure, which promotes actual orthogonality rather than only decorrelation. This distinction allows our approach to not only increase expert diversity but also better preserve pairwise geometric relationships between routed inputs, contributing to more stable routing behavior during training.

## 6    LIMITATIONS

While we train our models with relatively large data multipliers, prior work such as Muennighoff et al. (2025) suggests that substantially more data (trillions of tokens) may be necessary to achieve strong performance on downstream benchmarks. Nevertheless, our training setup provides sufficient scale to meaningfully compare the relative effectiveness of different balancing methods, which we supplement with statistical significance comparisons.

Finally, although our architectural choices align with recent MoE literature, our study is limited to a single set of design decisions. We leave the exploration of alternative configurations to future work. For instance, we do not investigate how token dropping might affect the performance of our balancing mechanism (instead focusing on higher-quality dropless models (Gale et al., 2022)), which could be a valuable direction for further analysis.

## 7    CONCLUSION

In this work, we introduced a novel load balancing mechanism for Mixture-of-Experts (MoE) models that consistently outperforms popular approaches across two scales. We also proposed efficient, scalable metrics for quantifying expert redundancy, and demonstrated that models with lower redundancy—as measured by our proposed metric and existing methods—exhibit improved parameter efficiency.

## 8    ETHICS STATEMENT

We adhere to the terms of service and respect all relevant licenses of software used. The environmental impact of our experiments are negligible compared to full-scale trillion-token LLM training, and we find an improvement in the efficiency of language models.

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

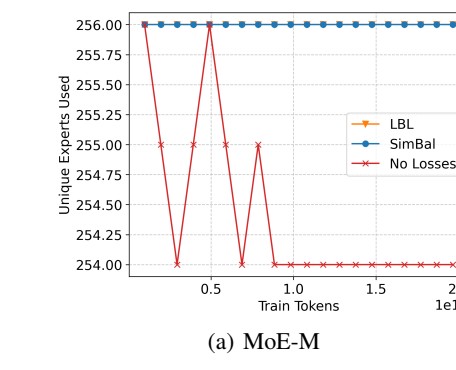 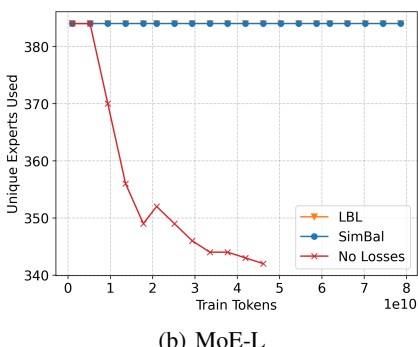

(a) MoE-M             (b) MoE-L

Figure 6: Expert utilization throughout training for MoE-M (left) and MoE-L (right), comparing LBL, our method (SimBal), and a baseline with no load balancing. We measure the number of unique experts activated on our full 77M-token validation set over time. Without any balancing, the expert routing collapses to a smaller set of experts. Both LBL and SimBal maintain full expert avoid expert collapse. The no-loss baseline was truncated early.

# A  APPENDIX

## A.1  ICLR LARGE LANGUAGE MODEL USAGE

Large language models (LLMs) were used to assist in the writing of the paper, and all outputs were thoroughly vetted and edited prior to being used.

## A.2  LOSS-FREE LOAD BALANCING COMBINATION

Table 8: Model setup and performance. Sequence-wise Expert Utilization (SEU) is computed as the mean over the fraction of activated experts within a sequence. SIMBAL can improve sequence-wise balance without significant performance degradation, sometimes improving performance. All models use all experts throughout the full validation set, LF is the least balanced per-batch. While LBL asserts near-perfect balance, it also causes substantial perplexity degradation.

| Model | MoE-M | MoE-M | MoE-M | MoE-M | MoE-M | MoE-M |
|---|---|---|---|---|---|---|
| **Gating** | Softmax | Softmax | Softmax | Sigmoid | Sigmoid | Sigmoid |
| **Balancing** | LF | LF+LBL | LF+SimBal | LF | LF+LBL | LF+SimBal |
| **Perplexity ↓** | 13.708 | 14.154 | 13.695 | 13.618 | 14.015 | 13.637 |
| **SEU ↑** | 0.505 | 0.997 | 0.755 | 0.381 | 0.997 | 0.476 |

Loss-Free (LF) balancing (Wang et al., 2024) applies a direct bias to routing scores ($s = xR$, rather than routing weights $r = G(xR)$) without adding an auxiliary loss. Let $f_i$ be the expert frequency in the current batch and $\bar{f} = 1/E$ the uniform target. Each expert's score is adjusted by a fixed scalar $\gamma$:

$$b'_i = b_i + \gamma \cdot \text{sign}(\bar{f} - f_i) \tag{7}$$

The scores are then used for computing the top-$A$ experts with the new scores $s_i$:

$$s_i = xR + b_i \tag{8}$$

This encourages uniform expert assignment, but is not used in the weighting of the experts ($r$). It thus allows non-uniform expert weighting but still allocates experts uniformly over the full dataset. Additionally, $\gamma$ is a hyperparameter that may need to be tuned, though the original authors recommend 0.001 since it provides a good balance between balancing while preventing fluctuations later in training.

Other work (DeepSeek-AI et al., 2025) use LBL in conjunction with LF for batch-wise load balancing, as they find that it can result in substantial imbalance in expert use sequence-wise. We do not include

these results in earlier charts due to this extreme imbalance. Instead, in this section, we explore whether a combination with SIMBAL works similarly to LBL to improve sequence-wise balancing.

While the original authors of LF use sigmoid gating (over our softmax gating), we find that softmax gating is substantially more common in state-of-the-art work. Thus, to maximize relevance (regardless of performance), we additionally compare with softmax gating. The training setup for MoE-M remains identical to Section 3.2 otherwise.

We evaluated the balancing capabilities of this method using the MoE-M configuration, comparing its performance against both LBL and SIMBAL. We summarize our results in Table 8. We find that sigmoid gating leads to significant degradation in sequence-wise balance, especially compared to using only SIMBAL or LBL (as seen in Table 4). In exchange, there was a minor and possibly statistically insignificant (using the deviation values from Section 4.2. This is not ideal, as with larger models, when using model parallelism, extra consideration may be needed to ensure full utilization of all devices. Using LBL mitigates some of this, but leads to a substantial degradation in performance.

## A.3 LAYER-WISE ORTHOGONALIZATION

We provide tables for layer-wise orthogonalization performance for SIMBAL, and compare the results to LBL on MoE-M (Table 9) and MoE-L (Table 10). LBL alone does not orthogonalize the router whatsoever, while SIMBAL is able to achieve mean squared error similar to commonly used $\epsilon$ for numerical stability.

| Router | SimBal | LBL |
|---|---|---|
| Layer 0 Router | 1.94017e-10 | 0.00146701 |
| Layer 1 Router | 1.70156e-10 | 0.01486 |
| Layer 2 Router | 1.91267e-10 | 0.0155954 |
| Layer 3 Router | 1.89254e-10 | 0.0102319 |
| Layer 4 Router | 1.50925e-08 | 0.0100937 |
| Layer 5 Router | 2.99727e-08 | 0.0143029 |
| Layer 6 Router | 1.82301e-10 | 0.020765 |
| Layer 7 Router | 1.73648e-10 | 0.0258847 |

Table 9: Router orthogonality of MoE-M, as measured by $(R^T R - I)^2$

| Router | SimBal | LBL |
|---|---|---|
| Layer 0 Router | 1.49951e-08 | 0.0125956 |
| Layer 1 Router | 1.00854e-10 | 0.027788 |
| Layer 2 Router | 1.03228e-10 | 0.0183506 |
| Layer 3 Router | 4.47955e-08 | 0.0128958 |
| Layer 4 Router | 1.5001e-08 | 0.00668315 |
| Layer 5 Router | 9.38376e-11 | 0.00399825 |
| Layer 6 Router | 1.16159e-10 | 0.00375414 |
| Layer 7 Router | 2.99078e-08 | 0.00736187 |
| Layer 8 Router | 4.47949e-08 | 0.0200508 |
| Layer 9 Router | 2.99088e-08 | 0.0377724 |
| Layer 10 Router | 5.97087e-08 | 0.083971 |
| Layer 11 Router | 1.49907e-08 | 0.138501 |

Table 10: Router orthogonality of MoE-L, as measured by $(R^T R - I)^2$

## A.4 IMPLEMENTATION DETAILS

Here we provide some implementation details related to the auxiliary losses used in the paper in Figure 7. For our LBL baseline, we use an open-source repository implementation based on Zoph et al. (2022), available at lucidrains/st-moe-pytorch on GitHub. For both, we multiply the output of the function by the scaling coefficient if/where applicable during training. These losses can then be added to the final model loss (by adding them), or included using the AddAuxiliaryLoss autograd trick used in DeepSeek's modeling_deepseek.py on HuggingFace.

```python
import torch
from einops import reduce

# LBL
def balance_loss(gates: torch.Tensor) -> torch.Tensor:
    batch_size, num_tokens, num_experts = gates.shape

    # bal_loss = E * sum(f_i * P_i), expert i
    # impl largely stolen from lucidrains/st-moe-pytorch
    # compatible with sigmoid or softmax gating
    expert_mask = gates > 0.0
    f_i = reduce(expert_mask.float(), "b t e -> b e", "mean")
    P_i = reduce(gates, "b t e -> b e", "mean")
    loss_per_batch = num_experts * torch.sum(f_i * P_i, dim=-1)
    return loss_per_batch.mean()

# SimBal
def simbal_loss(router_linear, p=1):
    w = router_linear.weight
    # no transpose needed since w is assumed to be the router
    # in jax w should be transposed due
    # to linear implementation differences
    # thus w.shape[0] <<< w.shape[1]
    w_ortho = torch.matmul(w, w.T)
    eye = torch.eye(w.shape[0], device=w.device)
    loss = torch.norm(w_ortho - eye, p=p)
    return loss
```

Figure 7: Python implementations of the LBL and SimBal loss functions.

