# OpenReview forum: "Load Balancing Mixture of Experts with Similarity Preserving Routers"
_ICLR.cc/2026/Conference — Submitted to ICLR 2026_

### Official Review · Reviewer_JVb3 · 2025-10-30

**Soundness:** 2
**Presentation:** 3
**Contribution:** 1
**Rating:** 4
**Confidence:** 3

**Summary:**

This paper proposed SIMBAL which uses a router orthogonalization loss that is believed to encourage expert specialization. The authors claim significant improvements over the traditional auxiliary load balancing loss, and claim that it removes balancing hyper-parameters completely.

**Strengths:**

1. The authors claim significant improvements over the traditional auxiliary load balancing loss.
2. The authors claim that it removes balancing hyper-parameters completely.
3. The author provides much analysis.

**Weaknesses:**

1. The central idea of an orthogonality loss on the router weights is conceptually similar to the "router orthogonalization loss" in the ERNIE 4.5 technical report. It would be beneficial to cite this parallel and frame the work's distinct contributions.
2. The orthogonality of router does not necessarily leads to different expert routing behavior. Suppose R=BQ（Q^TQ = I_E, B\in R^{DxD}）, one can easily apply B^{-1} to input x (e.g. absorb into the former layers' mlp), so that any router's behaviour equals to an orthogonal router on a routed input.
3. I doubt the gain comes from the scale constraint rather than the constraint on direction. It may affect the logits size so as to affect the loss.

**Questions:**

1. I suggest some experiments may be done to address point3 in weakness.
2. Explanation on point2 in weakness is welcome.

---

> ### Author Response · Authors · 2025-11-19
>
> Thank you for your thoughtful review. We address your points below.
>
> **Regarding the differences to ERNIE 4.5:** we were not aware of this parallel work and agree that it is valuable to acknowledge it. While ERNIE 4.5 also considers orthogonality in the router, there are several important differences in implementation. For instance, they normalize features within the router before computing the Gram matrix and use an L2 loss, whereas our approach directly encourages the router weights to approximate an orthogonal matrix using an L1 loss. The subtle detail of ERNIE normalizing the router weights for their loss is important: their loss measures cosine similarity to the identity, which does not encourage orthogonality (in the sense that R^T R = I), whereas our formulation directly promotes R^T R \approx I. As a result, our loss not only encourages expert specialization but also better preserves pairwise geometric relationships between the routed inputs, which improves routing stability during pretraining. In contrast, the ERNIE 4.5 loss primarily decorrelates expert partitions without directly encouraging orthogonality or pairwise preservation. We have cited ERNIE in the revised paper and have noted the differences in the related work.
>
> **Regarding the effect of orthogonality on routing behavior:** while it is theoretically possible to absorb a rotation into earlier layers in the scenario you stated, in our scenario the nonlinearities such as the softmax gating function of the router or SwiGLU of the previous layers should  prevent exact equivalence. To show that routing behavior has changed drastically between the two, we attach a routing saturation chart in Figure 1 of this reply (and Figure 5 of the paper), which measures agreement in expert assignments between the final checkpoint and prior checkpoints. We find that SimBal reaches saturation quickly and exhibits slower changes in expert usage later in training for both MoE-L and MoE-M. Together, these observations indicate that the orthogonality constraint has a significant impact on the routing behavior during training.
>
> **On the effect of logit size versus direction:** we can separate these effects by comparing the difference between the pairwise cosine similarity and the mean norm of the logits after training. We find that the mean norm of the outputs from the linear transformation of the SimBal router is 5.7070, compared to 6.2031 for LBL, while the corresponding angular distortions are 0.1686 and 0.3431, respectively. This indicates that the difference in logit magnitude is relatively small, whereas the improvement in directional alignment is substantial, suggesting that the benefits of our orthogonality-based loss primarily arise from better preservation of angular relationships.
>
> We greatly appreciate your careful reading of the paper. If any part would benefit from further clarification or revision, we would be happy to expand on it.
>
> [Figure 1] Router saturation chart. https://imgur.com/a/scdaAHj

---

### Official Review · Reviewer_D18y · 2025-10-31

**Soundness:** 2
**Presentation:** 2
**Contribution:** 1
**Rating:** 2
**Confidence:** 3

**Summary:**

This paper addresses the problem of redundancy and inconsistent routing in MoE models. The authors argue that standard load-balancing losses, which enforce a uniform distribution of tokens to experts, are inefficient. To solve this, they propose SIMBAL, that encourages the router's weight matrix (R) to be orthogonal by penalizing the difference between its Gram matrix (R^T R) and the identity matrix (I_E).

**Strengths:**

The idea of "similar tokens should be routed similarly" to preserve semantic consistency, is interesting. However, the methodology and computation cost are questionable.

**Weaknesses:**

Please see the question block.

**Questions:**

As mentioned in the abstract and throughout the paper: the goal of this model is to preserve the pairwise angles of the inputs (this means if two input $h_1$ and $h_2$ are similar, their routing should also be similar). This is achieved by promoting orthogonality in the router weights, because orthogonal matrices are dot-product ... preserving ....

However, preserving the dot product of the D_M-dimensional inputs require the router R to satisfy R R^T = I_{D_M}.  But, this seems weird for me, as R is a  $D_M \gg E$ matrix, (where the input dimension D_M is much larger than the number of experts E). Also the proposed loss  ||R^T R - I_E||_1, is a standard regularization that enforces orthogonality on the columns of $R$, encouraging diversity among experts.  So, it’s not clear how enforcing this diversity aligns with the goal of preserving input similarity. I think there is some mismatch and needs clarification.

The paper says the proposed PES requires less computation and is highly scalable. again in the same paragraph states that calculating PES requires inference once with the full model parameter, through all experts (a multiplier of 3.6-4.9x FLOPs in our case). Indeed,  4-5x increase in FLOPs is not cheap or scalable. Running all experts for a single token is computationally expensive.

PES (Eq. 6) measures cos(f_i(x), f_j(x)), the similarity between the outputs of all $N$ experts for the same token $x$. But again, this requires dense computation for all experts for every input.

Optional suggestion: In table 5, the baselines used for comparison are from 2018-2019. I understand the paper aims to improve upon earlier models, but incorporating recent baselines for the comparison can provide a meaningful evaluation.

---

> ### Author Response · Authors · 2025-11-19
>
> Thank you for your review. We appreciate the opportunity to clarify several points and have revised our explanations and made relevant changes to the paper accordingly.
>
> **Regarding the connection between orthogonality and angle preservation:** you are absolutely correct that exact dot-product preservation in the original D_M​-dimensional space would require row-orthonormality rather than column-orthonormality of R, this is impossible when projecting from a higher to a lower-dimensional space. Our router is trained to be *tall semi-orthogonal*, meaning its columns form an orthonormal basis. This structure preserves angular relationships within the column space, and equivalently, preserves the angles between the orthogonal projections of the original D_M​-dimensional inputs onto that space. From this perspective, encouraging column diversity and preserving angular relationships are aligned objectives rather than contradictory ones given the structure of the router. To support this empirically, we computed the L1 (as it is more human-interpretable)​ distortion in pairwise cosine similarities across 1 million tokens. We find that LBL’s router yields an average distortion of 0.3431, whereas SimBal reduces this to 0.1686, showing that our approach substantially improves the preservation of input pairwise relationships.
>
> **Regarding PES scalability:** Generally, the 4-5x overhead of a PES pass relative to a normal MoE forward is negligible for training because PES is an evaluation metric and is computed only occasionally. The training throughput difference between SimBal and LBL is therefore insignificant. We use PES because it is far more computationally efficient than traditional expert dropping, which *would* have a significant overhead during training due to the extreme number of comparisons required to parse the information we needed for our overlap analysis.
>
> Expert dropping measures redundancy by removing or replacing experts and observing the resulting performance change. To identify redundant experts across all layers (not just among the top activated ones), we must ablate each lower-weighted expert and compare its effect against that of the lowest-weighted active expert. Since expert assignments in deeper layers depend on outputs from earlier layers, these ablations cannot be consolidated into a single pass. This results in (E - E_active) * N_layers​ forward passes, which over the 8 MoE-M or 12 MoE-L layers leads to *hundreds* (224-336x) of full evaluations even with caching. In practice, this is extremely costly relative to training and makes expert dropping unsuitable for routine use.
>
> PES, in contrast, requires only E / E_active​ times the per-layer expert computation and depends solely on same-layer expert prediction overlap, allowing it to be computed in one pass. Because the rest of the model (e.g., attention) does not incur extra cost, the actual overhead is only 3.5-4.9x (vs. the original 224-336x). PES thus provides a continuous and fine-grained redundancy signal while being orders of magnitude cheaper than expert dropping. We have clarified this in Section 3.3 as the motivation for PES.
>
> **Regarding comparisons:** We selected LBL because it remains widely used in recent literature as the primary auxiliary balancing loss and therefore provides a common and relevant point of comparison. We did evaluate a more recent approach (loss-free balancing [1], mentioned in Section 5), but due to instability with the load balancing in our setting, we ultimately moved a thorough analysis of those results to Appendix A.2. We experimented with several possible router setups, including one closer to what they utilized in their paper for fairness, but found that load balance is still heavily degraded. We find that it is possible to combine their mechanism with ours to help load balance the model while retaining strong performance (with a ~50% improvement in SEU, similar perplexity), unlike LBL where perplexity is greatly degraded despite near-perfect SEU.
>
> We greatly appreciate your careful reading of the paper. If any part would benefit from further clarification or revision, we would be happy to expand on it.
>
> [1] Wang et al, "Auxiliary-Loss-Free Load Balancing Strategy for Mixture-of-Experts," 2024.

---

### Official Review · Reviewer_H3da · 2025-11-01

**Soundness:** 4
**Presentation:** 3
**Contribution:** 4
**Rating:** 6
**Confidence:** 4

**Summary:**

The paper proposes SIMBAL, an auxiliary router loss for Mixture-of-Experts models that preserves token-wise relational structure by softly encouraging orthogonality of the router. The motivation is that angle-preserving routers produce more consistent expert choices for similar inputs, reducing redundancy and speeding training. The authors define a computationally cheap orthogonality loss, propose Pairwise Expert Similarity (PES) as a scalable metric of expert redundancy, and show empirical gains over the standard Load Balancing Loss on two model scales: faster convergence during training, lower PES, and better final perplexity and downstream benchmarks.

**Strengths:**

The idea is clear and intuitive. A code snippet is also provided in the appendix, showing the simplicity of the implementation. Moreover, the auxiliary loss hyperparameter is not sensitive to tuning, which makes this method easily integratable into existing architectures.

The empirical gains are strong: training convergence is significantly faster with SIMBAL loss, and the final perplexity and benchmark scores are also better for MoE trained with SIMBAL.

New metric for expert redundancy that quantifies the similarity of experts without requiring much computation

Evaluation on a diverse set of pretraining benchmarks, with consistent improvements across all of them.

**Weaknesses:**

The authors mention that stronger benchmark performance is realized when training on significantly more data than the datasets used in the paper. If possible, it would be good to see some results on how the SIMBAL method performs when training on these larger datasets, compared to traditional MoE.

The paper motivates orthogonality as angle preserving, but a more formal connection between router orthogonality and reduced redundancy / improved specialization (maybe via an analysis of routing variance or router saturation) would improve the justification of this approach.

The authors mention the loss calculation is cheap but it would be more convincing to see the comparison of training throughput between this method and standard MoE (if there is a difference).

**Questions:**

Can you compare the training throughput of standard MoE and SIMBAL? Is there a noticeable difference with SIMBAL loss or is it negligible?

How do the improvements depend on number of experts, and number of active experts? Does the performance change if the MoE is more or less sparse?

What happens if you combine SIMBAL and LBL? Can they work together or do they conflict with each other?

PES is computed over 4M sampled tokens, how does the measurement vary over different token samples? Can you report the variance across samples?

---

> ### Author Response · Authors · 2025-11-19
>
> Thank you for your review. We address your points below.
>
> **Training throughput:** We did not observe any meaningful throughput differences compared to standard LBL. In a small run on 2 A100 80GB GPUs (DDP, batch size 48 per device), both SimBal and LBL achieved 0.93 steps/second (on MoE-M). Inference-level results in Table 6 of the paper also show that by pruning experts under a small weight threshold, SimBal matches or exceeds LBL inference speed while still providing substantial quality improvements.
>
> **Impact of expert count and sparsity:** We expect SimBal to benefit increasingly from larger numbers of experts, since the dimension of the preserved space will be higher relative to the input vector, which may help with consistency. However, we do expect the difference to be smaller when there are very few experts for the same reason. In practice, many recent MoE works such as [1] use very granular experts (128 total, 4 active), so this is less of a concern.
>
> **Scaling to larger datasets:** Due to compute constraints, we were not able to evaluate SimBal on substantially larger datasets. However, our training uses a data multiplier larger than what scaling-law analyses (i.e., [2]) suggest is sufficient, and find that it is more than enough to appreciate the differences between the methods.
>
> **Router saturation and specialization:** To better illustrate the connection between SimBal and reduced redundancy, we include a router saturation plot (now added as Figure 5 in the paper, and attached as Figure 1 of this reply). SimBal reaches saturation significantly faster than LBL, indicating more stable routing over training. This aligns with our PES results: greater stability and token-wise structure preservation correlate with reduced expert redundancy and better specialization.
>
> **Combining SimBal and LBL:** While we have not directly tested combinations of the two losses, we believe they could work together in some settings. For example, in token dropping: SimBal’s loss additionally promotes decorrelated expert assignments, which may help mitigate redundancy introduced by LBL, while LBL can enforce the more uniform sequence-wise distribution over experts if needed for systems-level optimization. That said, some hyperparameter tuning would likely be needed to balance the two signals. We perform an analysis of combining SimBal with a different loss-free load balancing mechanism [3] in Section A.2 of the appendix and find that SimBal is able to improve on their load balancing when both are used.
>
> **Variance of PES measurements:** For the minimum PES values in Table 4 (MoE-L), the variance across batches is 0.000137 for SimBal and 0.00185 for LBL.
>
> We greatly appreciate your careful reading of the paper. If any part would benefit from further clarification or revision, we would be happy to expand on it.
>
> [1] OpenAI, "gpt-oss-120b & gpt-oss-20b Model Card," 2025.
>
> [2] Kaplan, et al, "Scaling Laws for Neural Language Models," 2020.
>
> [3] Wang et al, "Auxiliary-Loss-Free Load Balancing Strategy for Mixture-of-Experts," 2024.
>
> [Figure 1] Router saturation chart. https://imgur.com/a/scdaAHj

---

### Official Review · Reviewer_CzLF · 2025-11-02

**Soundness:** 4
**Presentation:** 4
**Contribution:** 3
**Rating:** 8
**Confidence:** 4

**Summary:**

This paper provides a simple strategy to regularize the MoE router to avoid some issues with load-balancing loss (LBL) which is commonly used. The LBL encourages uniform routing to prevent expert collapse and token dropping, but as a result may create redundancy and lack of specialization among the experts. Intuitively, it is desirable that the routing decisions are similar for similar inputs, but this may be suppressed by LBL. To explicitly encourage this, the authors attempt to constraint the router to be orthogonal via a soft-regularization penalty. This penalty is simply the l1 distance of the router gram matrix to the identity. Adding the penalty does not add significant overhead per step and is able to reach a given target loss in fewer steps than LBL.

**Strengths:**

The authors provide a simple and principled approach to regularizing an MoE router in order to promote expert regularization and mitigate the redundancies in traditional load balancing loss approaches. This appears to substantially speed up MoE training. The empirical evidence is thorough and convincing.

**Weaknesses:**

Minor: The term load-balancing loss for SIMBAL seems slightly incorrect.

**Questions:**

The observed balance (high SEU) appears to be emergent, and is not guaranteed. It’s not obvious why orthogonality prevents collapse, can the authors comment on this?

If per-token entropy decreases but SEU remains high (Table 3), what exactly is being balanced?

How helpful is this approach when using a shared expert?

Could excessive orthogonality reduce beneficial expert overlap?

---

> ### Author Response · Authors · 2025-11-19
>
> Thank you for your review. We address your questions below.
>
> 1. Intuitively, encouraging orthogonality in the router outputs limits the extent to which different tokens rely on the same small subset of experts. If two sets of expert assignments for disparate tokens become overly similar, the orthogonality would increase, gently steering the router toward more diverse assignments.
>
> 2. Per-token entropy measures how uniform each token’s expert probability distribution is, while SEU reflects the aggregate distribution of actual expert usage across the dataset. A decrease in per-token entropy suggests that tokens become more confident in their preferred experts, but SEU can remain high as long as different experts continue to be used with varied weightings. Only when entropy drops substantially would this typically indicate that the router is making overly concentrated predictions, at which point SEU would be expected to decrease as well. We show both to indicate that while our entropy is lower, it is not low enough to drastically degrade balance.
>
> 3. We have not yet evaluated the method in architectures that include a shared expert. However, since our loss operates only on the router (which routes to non-shared experts), it can likely be applied without modification. We anticipate similar overall behavior.
>
> 4. Regarding beneficial expert overlap, we agree that some degree of overlap could be helpful in certain out-of-distribution scenarios, where tokens erroneously routed to a less-suitable expert might gain from additional redundancy. However, the overall accuracy of the model in these cases tends to degrade significantly regardless of routing strategy, suggested by the top expert dropping analysis (Figure 3 in our paper) where misallocation of just the top 2 experts results in a nearly 3x increase in validation loss. While some overlap may offer marginal robustness in these edge cases (LBL degrades less in this scenario), the benefit is likely limited given the overall difficulty of the setting.
>
> We greatly appreciate your careful reading of the paper. If any part would benefit from further clarification or revision, we would be happy to expand on it.

---

> > ### Comment · Reviewer_CzLF · 2025-11-27
> >
> > Thank you for the response, I will maintain my positive evaluation.

---

### Comment · Area_Chair_pRuU · 2025-11-27
**Rebuttal and Discussion Phase**

Dear Reviewers,

Thank you again for your time and effort in reviewing this paper. We are approaching the discussion deadline. I kindly ask you to review the rebuttal and continue the discussion so that we can reach a well-considered decision.

---

### Meta-Review · Area_Chair_xapv · 2026-01-06

**Summary:**

Reviewers generally found the proposed alternative to standard load-balancing loss interesting and practically appealing, and acknowledged the empirical improvements in convergence and redundancy metrics. However, several reviewers raised fundamental concerns about the mechanistic justification of the approach. In particular, questions remain as to whether encouraging router orthogonality introduces a genuine inductive bias on routing behavior, or whether the observed gains could instead be explained by reparameterization effects or logit scaling, rather than the claimed preservation of token-wise similarity.

While the rebuttal provided thoughtful clarifications, additional diagnostics, and empirical evidence that routing behavior changes during training, reviewers’ core concerns about causal attribution of the gains were not fully resolved. Taken together with concerns about overlap with closely related orthogonalization approaches and the absence of more targeted ablations to isolate directional versus scale effects, these unresolved issues reduced confidence in the conceptual grounding of the method, informing the final recommendation.

**Reviewer Concerns:**

The rebuttal addressed several concerns, including clarifying the relationship to prior work (e.g., ERNIE 4.5), providing additional empirical analyses of routing dynamics. However, core mechanistic concerns remain outstanding, whether the orthogonality constraint induces a genuine and irreducible inductive bias on routing behavior, or whether the observed improvements can be explained by changes in logit scale. In addition, the rebuttal does not fully resolve the ambiguity between directional versus scale-based effects, as no targeted ablation is provided to isolate these factors.

**Reviewer Scores:**

see above

---

### Decision · Program_Chairs · 2026-01-26

Reject